# The Value of Information
# When Deciding What to Learn

**Dilip Arumugam**
Department of Computer Science
Stanford University
dilip@cs.stanford.edu

**Benjamin Van Roy**
Department of Electrical Engineering
Department of Management Science & Engineering
Stanford University
bvr@stanford.edu

## Abstract

All sequential decision-making agents explore so as to acquire knowledge about a particular target. It is often the responsibility of the agent designer to construct this target which, in rich and complex environments, constitutes a onerous burden; without full knowledge of the environment itself, a designer may forge a sub-optimal learning target that poorly balances the amount of information an agent must acquire to identify the target against the target's associated performance shortfall. While recent work has developed a connection between learning targets and rate-distortion theory to address this challenge and empower agents that decide what to learn in an automated fashion, the proposed algorithm does not optimally tackle the equally important challenge of efficient information acquisition. In this work, building upon the seminal design principle of information-directed sampling [Russo and Van Roy, 2014], we address this shortcoming directly to couple optimal information acquisition with the optimal design of learning targets. Along the way, we offer new insights into learning targets from the literature on rate-distortion theory before turning to empirical results that confirm the value of information when deciding what to learn.

## 1  Introduction

Information acquisition is a fundamental consideration in the design of sequential decision-making agents. Indeed, the literature over recent years has unequivocally demonstrated its theoretical [Russo and Van Roy, 2016, 2014, 2018a] and empirical [Osband et al., 2016a,b, Nikolov et al., 2018] importance in addressing the challenge of exploration in sequential decision-making problems. Despite this significance, a strategy for information acquisition, however efficient, is only as good as the information it uncovers. An agent that displays remarkable efficiency in seeking immaterial information offers little utility to its designer. Similarly, when faced with the rich complexity of real-world tasks, a computationally-bounded agent aspiring to learn that which is well beyond its constrained resources is equally ineffective. As a concrete example of the latter scenario, consider learning the optimal policy for the game of Go [Silver et al., 2017] solely relying on the hardware of a mobile device. While not an entirely infeasible task, the amount of time needed to complete such an endeavor may force the agent designer into settling for a small fraction of the optimal policy performance in exchange for a substantial reduction in the difficulty of the learning problem.

Recent work has developed a connection between rate-distortion theory [Shannon, 1959, Berger, 1971] and the problem of deciding what an agent should learn [Arumugam and Van Roy, 2021], optimally balancing between the requisite information needed by any agent to identify a learning target and the corresponding performance shortfall associated with this target. Intuitively, the complexity of a learning problem can be quantified by the bits of information an agent must acquire from the

35th Conference on Neural Information Processing Systems (NeurIPS 2021).

environment in order to realize a particular target. Rate-distortion theory offers a principled framework for reasoning about a learning target that comprises the minimum number of bits from the environment while retaining some degree of near-optimality. Arumugam and Van Roy [2021] operationalize the classic Blahut-Arimoto algorithm [Blahut, 1972, Arimoto, 1972] in the multi-armed bandit setting to facilitate computationally-tractable agents that are not only capable of computing such an optimal target at the start of learning but also gradually adapt this target over time as the agent's knowledge of the environment accumulates. The resulting Blahut-Arimoto Satisficing Thompson Sampling algorithm (BLASTS) then applies the well-known principle of probability matching in order to select actions [Thompson, 1933, Russo et al., 2018]. While BLASTS demonstrates the desired ability to accommodate an agent designer in recovering a wide spectrum of policies with varying levels of performance and learning efficiency, it is well known that the underlying probability matching principle of Thompson Sampling constitutes a sub-optimal exploration strategy [Russo and Van Roy, 2018a].

In this work, we combine the optimal computation of learning targets via rate-distortion theory with optimal information acquisition. To achieve the latter goal, we generalize the information-directed sampling (IDS) principle of Russo and Van Roy [2018a] so that the fundamental information ratio captures expected regret and information gain about a learning target computed via the Blahut-Arimoto algorithm. As with BLASTS, the resulting algorithms which instantiate this Blahut-Arimoto Information-Directed Sampling principle (BLAIDS) retain a single, real-valued hyperparameter through which an agent designer expresses a preference for the balance struck by the resulting learning target. Moreover, just as BLASTS offers a generalization of Thompson Sampling capable of more efficiently computing optimal policies, we find that a particular computationally-tractable instantiation of BLAIDS mirrors this property relative to its IDS counterpart. En route to deriving BLAIDS, we outline the precise details of the Blahut-Arimoto algorithm itself and connect our setting with various results from the information-theory literature.

The paper proceeds as follows: in Section 2 we offer a general presentation of information-theoretic learning targets for sequential decision-making problems. We then hone in on the multi-armed bandit setting in Section 3 to carefully study incorporating the value of information when acting in pursuit of an arbitrary learning target. We conclude in Section 4 with a corroborating set of computational experiments that demonstrate the efficacy of our approach. Due to space constraints, we relegate background material on information theory, related work, and all technical proofs to the appendix.

## 2 Learning Targets

This section focuses on a general treatment of learning targets via rate-distortion theory, moving beyond the assumptions of Arumugam and Van Roy [2021] and offering further insight into the mathematical details of the Blahut-Arimoto algorithm.

### 2.1 Problem Formulation

In this section, we follow the general problem formulation of Lu et al. [2021] that offers a unified treatment of sequential decision-making problems, spanning multi-armed bandits and reinforcement-learning problems [Lattimore and Szepesvári, 2020, Sutton and Barto, 1998]. We adopt a generic agent-environment interface wherein, at each time period, the agent executes an action $A_t \in \mathcal{A}$ within an environment $\mathcal{E}$ which results in an associated next observation $O_{t+1} \in \mathcal{O}$. This sequential interaction between agent and environment results in an associated history at each timestep $H_t = (A_0, O_1, \ldots, A_{t-1}, O_t) \in \mathcal{H}$ representing the action-observation sequence available to the agent upon making its selection of $A_t$. More formally, we may characterize the overall environment as $\mathcal{E} = \langle \mathcal{A}, \mathcal{O}, \rho \rangle$ containing the action set $\mathcal{A}$, observation set $\mathcal{O}$, and the observation function $\rho : \mathcal{H} \times \mathcal{A} \to \Delta(\mathcal{O})$ that prescribes the distribution over next observations given the current history and action selection:

$$\mathbb{P}(O_{t+1}|\mathcal{E}, H_t, A_t) = \mathbb{P}(O_{t+1}|H_t, A_t) \qquad O_{t+1} \sim \rho(\cdot|H_t, A_t).$$

An agent's policy $\pi : \mathcal{H} \to \Delta(\mathcal{A})$ encapsulates the relationship between the histories encountered in each timestep $H_t$ and the executed action $A_t$. We maintain that actions are independent of the environment conditioned on history, $A_t \perp \mathcal{E}|H_t$ such that $\pi(a|H_t) = \mathbb{P}(A_t = a|H_t)$ assigns a probability to each action $a \in \mathcal{A}$ given the history.

An agent designer expresses preferences across histories via a reward function $r : \mathcal{H} \times \mathcal{A} \times \mathcal{O} \to \mathbb{R}$ such that an agent enjoys a reward $R_{t+1} = r(H_t, A_t, O_{t+1})$ at each timestep. Given any finite time horizon $T < \infty$, the accumulation of rewards provide a notion of return $\sum_{t=0}^{T-1} R_{t+1}$. To develop preferences over behaviors, it is natural to associate with each policy $\pi$ a corresponding expected return or value across the horizon $T$ as $\overline{V}^\pi = \mathbb{E}\left[ \sum_{t=0}^{T-1} R_{t+1} \mid \mathcal{E} \right]$, where the expectation integrates over the randomness in the policy $\pi$ as well as the observation function $\rho$. A trademark across the sequential decision-making literature is the design of agents that strive to achieve the optimal value within the confines of some policy class $\Pi \subseteq \{\pi : \mathcal{H} \to \Delta(\mathcal{A})\}$, $\overline{V}^\star = \sup_{\pi \in \Pi} \overline{V}^\pi$. Notice that when rewards and the distribution of the next observation $O_{t+1}$ depend only on the current observation $O_t$, rather than the full history $H_t$, we recover the traditional Markov Decision Process [Bellman, 1957, Puterman, 1994] studied throughout the reinforcement-learning literature [Sutton and Barto, 1998]. Alternatively, when these quantities rely solely upon the most recent action $A_t$, we recover the traditional multi-armed bandit [Lattimore and Szepesvári, 2020].

## 2.2 The Curse of Curiosity

While the design of agents in pursuit of optimal policies is a perfectly natural object of study, it can often occur without regard for the complexity of the environment. In particular, the optimal policy is often a deterministic function of the environment such that, if an agent manages to identify the environment, it possesses all requisite information to determine the optimal policy. An agent designer reflects their initial uncertainty about the environment through a prior distribution $\mathbb{P}(\mathcal{E} \in \cdot)$; as the history unfolds, what can be learned from the agents current knowledge of the environment is represented by conditional probabilities $\mathbb{P}(\mathcal{E} \in \cdot | H_t)$. Through this prior distribution, a total of $\mathbb{H}(\mathcal{E})$ bits quantify all of the information needed for identifying the environment. Prior work has demonstrated both theoretically and empirically how it is an agent's incremental resolution of epistemic uncertainty over the environment $\mathcal{E}$ that gives rise to the efficient acquisition of these $\mathbb{H}(\mathcal{E})$ bits [Chapelle and Li, 2011, Russo and Van Roy, 2016, Osband et al., 2016a, Agrawal and Jia, 2017, O'Donoghue et al., 2018, Osband et al., 2019]. For sufficiently rich and complex environments, however, $\mathbb{H}(\mathcal{E})$ can become prohibitively large or even infinite, making the pursuit of an optimal policy entirely intractable.

This issue warrants the construction of a more general mechanism whereby an agent designer may express tractable learning targets, alternatives to the optimal solution that, while potentially incurring some degree of suboptimality, fall within the means of a computationally-bounded agent interacting with a highly-complex environment. Lu et al. [2021] introduce this notion of a learning target $\chi$ as a generic random variable and (possibly stochastic) function of the environment $\mathcal{E}$. Their proposed desiderata for the construction of learning target is two-fold: (1) demanding only a feasible amount of information from the environment $\mathbb{I}(\mathcal{E}; \chi)$ in order to reconcile $\chi$ and (2) incurring a bounded degree of performance shortfall by aiming for $\chi$, $\mathbb{E}[\overline{V}^\star - \overline{V}^{\pi_\chi}]$, where $\pi_\chi$ is a target policy induced by $\chi$. Observe, however, that this treatment of a learning target assumes its provision to the agent before interaction with the environment has begun. Consequently, the burden of specifying a suitable learning target that best balances between the two desiderata falls upon the agent designer. To remedy this, we begin by extending the formulation of Arumugam and Van Roy [2021] and emphasizing rate-distortion theory as a general tool for the design of optimal learning targets.

It is important to re-iterate and clarify that the role of $\pi_\chi$ is not to achieve performance that is on par with the optimal $\pi^\star$, but rather to strike an appropriate balance between performance shortfall relative to $\pi^\star$ and the ease of learnability; the latter desideratum is accurately captured mathematically via bits of information. As an agent shifts its focus away from $\pi^\star$ towards a $\chi$ that is easier to learn (in the information-theoretic sense used throughout this paper), we anticipate greater data efficiency. That is, an agent should be able to more quickly identify the desired $\chi$ relative to the requisite amount of time and data needed to identify $\pi^\star$. Naturally, in a complex environment, we expect this data efficiency to be critical and to come at some cost: the sub-optimality of the resulting solution (which we take as the distortion in our rate-distortion framing). An agent designer's willingness to tolerate this sub-optimality stems from the complexity of the environment and an acknowledgement of the potentially intractable amount of data needed to otherwise identify and recover $\pi^\star$. The more

performance shortfall one is willing to tolerate relative to optimal behavior, the more efficiently an agent should be able to find such a satisficing solution. Russo and Van Roy [2018b] already highlight one example of this phenomenon in the multi-armed bandit setting, where the presence of multiple arms can make pursuit of an $\varepsilon$-optimal arm far more data efficient than the optimal arm. From the perspective of the agent designer, one should not incorporate a learning target with the expectation of being competitive with the optimal policy, but rather with the goal of being more data efficient in the acquisition of the learning target relative to that of the optimal policy.

## 2.3 The Rate-Distortion Theory of Learning Targets

Rate-distortion theory is a sub-area of information theory that encapsulates the foundations of lossy compression [Shannon, 1959, Berger, 1971, Cover and Thomas, 2012]. Abstractly, rate-distortion theory formulates a lossy compression problem as a constrained optimization in the space of probability kernels or channels, given an information source and a tolerable upper threshold on distortion. The natural goal is to identify a channel that preserves the minimum number of bits from the information source while adhering to the specified distortion upper bound.

More formally, let $(\Omega, \mathcal{F}, \mathbb{P})$ be a probability space and consider a random variable $X : \Omega \to \mathcal{X}$ taking values on the measurable space $(\mathcal{X}, \mathbb{X})$ with an associated marginal distribution $\mathbb{P}_X(\cdot) = \mathbb{P}(X^{-1}(\cdot))$ that represents an information source. Similarly, define the random variable $\widehat{X} : \Omega \to \widehat{\mathcal{X}}$ that takes values on $(\widehat{\mathcal{X}}, \widehat{\mathbb{X}})$ and corresponds to a channel output. Given a known, measurable distortion function $d : \mathcal{X} \times \widehat{\mathcal{X}} \mapsto \mathbb{R}_{\geq 0}$ and a desired upper bound on distortion $D$, the rate-distortion function is defined as $\mathcal{R}(D) = \inf_{\mathbb{P}_{X, \widehat{X}} \in \Lambda} \mathcal{I}(X; \widehat{X})$ quantifying the minimum number of bits (on average) that must be communicated from $X$ across a channel in order to adhere to the specified expected distortion threshold $D$. Here, the infimum is taken over $\Lambda = \{\mathbb{P}_{X, \widehat{X}} \mid \forall A \in \mathbb{X} : \mathbb{P}_{X, \widehat{X}}(A \times \widehat{\mathcal{X}}) = \mathbb{P}_X(A)$ and $\mathbb{E}[d(X, \widehat{X})] \leq D\}$ representing the set of all joint distributions on $(X \times \widehat{X}, \mathbb{X} \times \widehat{\mathbb{X}})$ whose corresponding marginal distribution on $X$ matches the original information source $\mathbb{P}_X$ while also satisfying the constraint on bounded expected distortion. Intuitively, a higher rate corresponds to requiring more bits of information and smaller information loss between $X$ and $\widehat{X}$, enabling higher-fidelity reconstruction (lower distortion); conversely, lower rates reflect more substantial information loss, potentially exceeding the tolerance on distortion $D$. The remainder of this section is devoted to making the mathematical details of this problem precise in our specific context of specifying learning targets. We encourage readers to consult the appendix for the precise details of our notation.

At each timestep, an agent maintains its epistemic uncertainty over the environment via a posterior distribution $\mathbb{P}(\mathcal{E}|H_t)$. Since the derivation is identical for all timesteps, we suppress time and history from the notation for now and let $\mathbb{P}_{\mathcal{E}}$ denote this distribution at any arbitrary timestep. We will take the environment $\mathcal{E}$ and learning target $\chi$ random variables to be associated with measurable spaces $(\Sigma, \mathbb{S})$ and $(\Xi, \mathbb{X})$, respectively. Let $\mathbb{P}_{\mathcal{E}, \chi}$ defined on $(\Sigma \times \Xi, \mathbb{S} \times \mathbb{X})$ be a joint distribution over environment-target pairs. Furthermore, let $d : \Sigma \times \Xi \to \mathbb{R}_{\geq 0}$ be a known, measurable function that quantifies a notion of loss or *distortion* that occurs by using a particular realization of the learning target within a specific realization of the environment. We may define the rate-distortion function as[1]

$$\mathcal{R}(D) = \inf_{\mathbb{P}_{\mathcal{E}, \chi} \in \Lambda : \mathbb{E}[d(\mathcal{E}, \chi)] \leq D} \mathbb{I}(\mathbb{P}_{\mathcal{E}, \chi}) \triangleq \inf_{\mathbb{P}_{\mathcal{E}, \chi} \in \Lambda : \mathbb{E}[d(\mathcal{E}, \chi)] \leq D} \mathbb{I}(\mathcal{E}; \chi) \tag{1}$$

$$\Lambda = \{\mathbb{P}_{\mathcal{E}, \chi} \mid \forall A \in \mathbb{S} : \mathbb{P}_{\mathcal{E}, \chi}(A \times \Xi) = \mathbb{P}_{\mathcal{E}}(A) \text{ and } \forall t \in [T] : \chi \perp H_t | \mathcal{E}\} \tag{2}$$

where the infimum is taken over the constrained set of joint probability measures $\Lambda$ such that (1) the $\mathcal{E}$ marginal of $\mathbb{P}_{\mathcal{E}, \chi}$ matches the given source distribution $\mathbb{P}_{\mathcal{E}}$, (2) the expected distortion does not exceed the given threshold $D \in \mathbb{R}_{>0}$, and (3) the target is conditionally independent of all histories given the environment. These first two conditions are typical in rate-distortion theory while the final condition simply ensures that no history can offer more information about a learning target than what is already provided by the environment. While the distortion function $d$ will eventually be defined so as to quantify the performance shortfall of our learning target, we proceed with the generic,

---

[1]Typically, $\mathcal{R}(D)$ is defined in terms of the mutual information, however the (equivalent) parameterization via the joint distribution $\mathbb{P}_{\mathcal{E}, \chi}$ will be useful in subsequent derivations.

unspecified distortion function for now and assume that $\mathcal{R}(D) \to 0$ as $D \to \infty$. Following from this assumption along with the convexity of mutual information and Gibb's inequality, we may recover the basic facts that $\mathcal{R}(D)$ is a non-negative, convex, and monotonically-decreasing function in its argument [Cover and Thomas, 2012].

The details of computing rate-distortion functions as presented by Arumugam and Van Roy [2021] rely on the classic derivation based on calculus and Lagrange multipliers, under the assumption that the information source and channel output are discrete random variables [Blahut, 1972]. In brief and continuing with the first example of this section, the discrete algorithm proceeds by initializing $p_0(\widehat{x} \mid x) = |\widehat{\mathcal{X}}|^{-1}, \forall \widehat{x} \in \widehat{\mathcal{X}}$ and then iterating the following two updates

$$q_k(\widehat{x}) = \sum_{x \in \mathcal{X}} p(x)p_{k-1}(\widehat{x} \mid x) \qquad p_{k+1}(\widehat{x} \mid x) \propto q_k(\widehat{x}) \exp\left(-\beta d(x, \widehat{x})\right),$$

where $p(x)$ denotes the information source and $\beta$ is a Lagrange multiplier. While discreteness carries computational conveniences for practical implementation, here we offer a more general presentation of the Blahut-Arimoto algorithm for arbitrary environments and learning targets. It is important to clarify that the subsequent derivations of this section are classic results in the information-theory literature [Csiszár, 1974, Gray, 2011] and do not represent novel theoretical contributions of this work; that said, we do take our contribution to be distilling such key results from the information-theory literature for broader consumption by the sequential decision-making community and, to that end, offer a full presentation for the benefit of readers.

For the purposes of handling this constrained optimization, we introduce the following objective with parameter $\beta \geq 0$:

$$\mathcal{F}(\beta) = \inf_{\mathbb{P}_{\mathcal{E},\chi} \in \Lambda} \left(\mathbb{I}(\mathbb{P}_{\mathcal{E},\chi}) + \beta \mathbb{E}\left[d(\mathcal{E}, \chi)\right]\right) = \inf_{\mathbb{P}_{\mathcal{E},\chi} \in \Lambda} \left(D_{\text{KL}}(\mathbb{P}_{\mathcal{E},\chi} || \mathbb{P}_{\mathcal{E}} \times \mathbb{P}_{\chi}) + \beta \mathbb{E}\left[d(\mathcal{E}, \chi)\right]\right)$$

As noted by Csiszár [1974], $\mathcal{F}(\beta)$ is the largest vertical-axis intercept of a line with slope $-\beta$ that includes no point above the rate-distortion curve. Moreover, the rate-distortion function can be expressed in terms of this new objective, sweeping over all possible values of $\beta$.

**Lemma 1** (Lemma 1.2 [Csiszár, 1974]).

$$\mathcal{R}(D) = \max_{\beta \geq 0} \mathcal{F}(\beta) - \beta D$$

*The value of $\beta$ that achieves this maximum is characterized as being associated with the distortion threshold $D$. Conversely, a joint distribution $\mathbb{P}_{\mathcal{E},\chi}$ that achieves the infimum of $\mathcal{F}(\beta)$ has an associate rate $\mathcal{R}(D) = \mathbb{I}(\mathbb{P}_{\mathcal{E},\chi})$ where $\beta$ is associated with $D = \mathbb{E}_{\mathbb{P}_{\mathcal{E},\chi}}[d(\mathcal{E}, \chi)] \triangleq \int d(\mathcal{E}, \chi) d\mathbb{P}_{\mathcal{E},\chi}(\mathcal{E}, \chi).$*

Lemma 1 clarifies an important relationship between the parameter $\beta$ and the rate-distortion curve, accommodating computation of the rate-distortion function upon specification of a particular $\beta$ values rather than a distortion upper bound $D$; in particular, each $\beta$ corresponds to the slope of a tangent line to the rate-distortion curve such that, given a value of $\beta$, the associated joint distribution that achieves the infimum in $\mathcal{F}(\beta)$ generates the point $(D, \mathcal{R}(D))$ on the curve. Per Blahut [1972], computing the infimum in $\mathcal{F}(\beta)$ is handled by an alternating optimization procedure that relies on the following functional:

$$\mathcal{J}(\mathbb{P}_{\mathcal{E},\chi}, \mathbb{Q}_{\chi}, \beta) = D_{\text{KL}}(\mathbb{P}_{\mathcal{E},\chi} || \mathbb{P}_{\mathcal{E}} \times \mathbb{Q}_{\chi}) + \beta \mathbb{E}_{\mathbb{P}_{\mathcal{E},\chi}}[d(\mathcal{E}, \chi)]$$

which replaces the marginal distribution over $\chi$ induced by $\mathbb{P}_{\mathcal{E},\chi}$ with an alternative distribution $\mathbb{Q}_{\chi}$. When $\mathcal{J}(\mathbb{P}_{\mathcal{E},\chi}, \mathbb{Q}_{\chi}, \beta)$ is finite, this implies that $\mathbb{P}_{\mathcal{E},\chi} \ll \mathbb{P}_{\mathcal{E}} \times \mathbb{Q}_{\chi}$; moreover, if $\mathbb{E}_{\mathbb{P}_{\mathcal{E},\chi}}[d(\mathcal{E}, \chi)]$ is finite, then $\mathbb{P}_{\mathcal{E}} \times \mathbb{Q}_{\chi}(\{\mathcal{E}, \chi | d(\mathcal{E}, \chi) < \infty\}) > 0$. Taken together, these facts imply that for any realization of the environment

$$\alpha_{\mathbb{Q}_{\chi}, \beta}(\mathcal{E}) = \left[\int \exp\left(-\beta d(\mathcal{E}, \chi)\right) d\mathbb{Q}_{\chi}(\chi)\right]^{-1} < \infty$$

holds $\mathbb{P}_{\mathcal{E}}$-almost surely. Thus, for any marginal over learning targets $\mathbb{Q}_{\chi}$, we may induce a new joint distribution $\mathbb{Q}_{\mathcal{E},\chi}$ whose corresponding Radon-Nikodym derivative with respect to $\mathbb{P}_{\mathcal{E}} \times \mathbb{Q}_{\chi}$ is given by

$$\frac{d\mathbb{Q}_{\mathcal{E},\chi}}{d\mathbb{P}_{\mathcal{E}} \times \mathbb{Q}_{\chi}} = \alpha_{\mathbb{Q}_{\chi}, \beta}(\mathcal{E}) \exp\left(-\beta d(\mathcal{E}, \chi)\right) \tag{3}$$

While the corresponding $\chi$ marginal of this new joint distribution need not be identical to $\mathbb{Q}_\chi$, we can show that our original source distribution $\mathbb{P}_\mathcal{E}$ is preserved under the corresponding $\mathcal{E}$ marginal. In particular, for any measurable set $A \in \mathbb{S}$, we have

$$\mathbb{Q}_{\mathcal{E},\chi}(A \times \Xi) = \int_{F \times \Xi} \frac{d\mathbb{Q}_{\mathcal{E},\chi}}{d\mathbb{P}_\mathcal{E} \times \mathbb{Q}_\chi} d\mathbb{P}_\mathcal{E} \times \mathbb{Q}_\chi(\mathcal{E}, \chi) = \int_{F \times \Xi} \alpha_{\mathbb{Q}_\chi, \beta}(\mathcal{E}) \exp\left(-\beta d(\mathcal{E}, \chi)\right) d\mathbb{P}_\mathcal{E} \times \mathbb{Q}_\chi(\mathcal{E}, \chi)$$

$$= \int_A \underbrace{\left(\alpha_{\mathbb{Q}_\chi, \beta}(\mathcal{E}) \int \exp\left(-\beta d(\mathcal{E}, \chi)\right) d\mathbb{Q}_\chi(\chi)\right)}_{=1} d\mathbb{P}_\mathcal{E}(\mathcal{E}) = \mathbb{P}_\mathcal{E}(A),$$

where the second line uses the aforementioned integrability conditions to apply Fubini's Theorem. To recover an alternating optimization procedure for minimizing our objective and computing the rate-distortion function, we may leverage the following relationship between the original and new joint distributions:

**Lemma 2** (Lemma 1.2 [Csiszár, 1974]). *Let $\mathbb{Q}_\chi$ be an arbitrary marginal distribution over $\chi$. Moreover, let $\mathbb{P}_{\mathcal{E},\chi}$ be an arbitrary joint distribution over $\mathcal{E}, \chi$ with an associated marginal distribution $\mathbb{P}_\chi$ and take $\mathbb{Q}_{\mathcal{E},\chi}$ to be the joint distribution as defined in Equation 3. Then,*

$$\mathcal{J}(\mathbb{P}_{\mathcal{E},\chi}, \mathbb{Q}_\chi, \beta) \geq \mathcal{J}(\mathbb{P}_{\mathcal{E},\chi}, \mathbb{P}_\chi, \beta)$$
$$\mathcal{J}(\mathbb{P}_{\mathcal{E},\chi}, \mathbb{Q}_\chi, \beta) \geq \mathcal{J}(\mathbb{Q}_{\mathcal{E},\chi}, \mathbb{Q}_\chi, \beta)$$

Lemma 2 prescribes a natural algorithm for the computation of rate-distortion functions. Upon the provision of a source distribution over environments $\mathbb{P}_\mathcal{E}$ and an initial marginal distribution over learning targets $\mathbb{Q}_\chi^{(1)}$, one may first compute an updated channel distribution $\mathbb{Q}_{\mathcal{E},\chi}^{(1)}$ per Equation 3. The induced marginal of $\mathbb{Q}_{\mathcal{E},\chi}^{(1)}$ serves as the next distribution $\mathbb{Q}_\chi^{(2)}$, giving rise to another updated channel $\mathbb{Q}_{\mathcal{E},\chi}^{(2)}$. Naturally, this process may continue iteratively until convergence. By the second inequality of Lemma 2, the updated channel distributions $\mathbb{Q}_{\mathcal{E},\chi}^{(k)}$ are non-increasing in the objective and, analogously, the induced marginal distribution is also guaranteed to not increase the objective by the first inequality. This procedure is precisely the classic Blahut-Arimoto algorithm [Blahut, 1972, Arimoto, 1972], generalized beyond the assumptions of discrete source and channel output random variables.

## 2.4 Target Policies for Reinforcement Learning

With the results of the previous section in hand, an agent may, at any timestep, run the Blahut-Arimoto algorithm using its current posterior beliefs over the environment $\mathbb{P}(\mathcal{E}|H_t)$ as a source distribution in order to compute a learning target that optimally negotiates between the information required to learn and the resulting target performance shortfall. In the context of reinforcement-learning problems, this learned target $\chi$ corresponds to a particular target policy $\pi_\chi$ and, to ensure the latter criterion holds, a natural notion of distortion is the expected squared regret between the optimal and target policies:

$$d(\mathcal{E}, \chi | H_t) = \mathbb{E}\left[(\overline{V}^\star - \overline{V}^{\pi_\chi})^2 | \mathcal{E}, H_t\right] = \mathbb{E}\left[\left(\mathbb{E}_{\pi_\chi}\left[\sum_{t'=t}^{T-1} (V^\star(H_{t'}) - Q^\star(H_{t'}, A_t))\right]\right)^2 \bigg| \mathcal{E}, H_t\right].$$

The final equality follows from a variant of the performance-difference lemma [Kakade and Langford, 2002] and is proven for our setting as Theorem 1 of Lu et al. [2021]. Unfortunately, while the results of the previous section hold for this particular sequence of rate-distortion functions, the choice of $\chi$ as a target policy represents an obstacle to practical implementation. In particular, this would imply that the resulting channel of the Blahut-Arimoto algorithm would map between realizations of the environment $\mathcal{E}$ to target policies $\pi : \mathcal{H} \to \Delta(\mathcal{A})$. Moreover, computation of the distortion associated with each possible realization of the target policy $\pi_\chi$ would require either an independent policy-evaluation step to compute the value function $\overline{V}^{\pi_\chi}$ or demand sampling trajectories of each potential $\pi_\chi$ from the environment in order to leverage the final performance shortfall decomposition shown above.

One possible resolution to these issues might include leveraging recent progress in unsupervised skill discovery [Eysenbach et al., 2018] such that potential target policies $\pi_\chi$ all come from the resulting parameterized policy class. Computing the associated universal value function approximator [Schaul et al., 2015] for this policy class would allow for efficient computation of the distortion function during the rate-distortion optimization. As an alternative, one might further restrict focus to a tabular Markov Decision Process, at which point the target policy in question can be expressed as a finite sequence of action random variables (one per state); it may then be possible to "stitch" $\pi_\chi$ from the solutions to each of the state-dependent rate-distortion functions.

For now, we leave the question of how to adaptively compute optimal target policies in a computationally-tractable manner to future work and, instead, turn our attention to the multi-armed bandit setting where we may more feasibly study the coupling of optimal learning targets with optimal information acquisition.

## 3 Target Actions & The Value of Information

In order to develop a computationally-tractable algorithm for combining learning targets with IDS, this section focuses on multi-armed bandit problems where we take the learning target $\chi$ to be a target action the agent aims to identify from the environment.

### 3.1 Blahut-Arimoto Satisficing Thompson Sampling

We begin by taking a closer look at the design of BLASTS and examine the extent to which learning targets computed via the Blahut-Arimoto algorithm are preferable to hand-crafted learning targets. As a first observation, we recall that while BLASTS does employ the Blahut-Arimoto algorithm as described in the previous section, the function computed by Arumugam and Van Roy [2021] is actually the so-called "plug-in" estimator for the rate-distortion function [Harrison and Kontoyiannis, 2008]; rather than directly using the agent's representation of environment uncertainty as an information source, BLASTS draws a fixed number of posterior samples $z$ and uses the resulting empirical distribution as an input source to the Blahut-Arimoto algorithm. Such an approach is amenable to scenarios where an agent is only able to sample from its beliefs over the environment but cannot reliably provide estimates of likelihoods [Osband et al., 2016a, Lu and Van Roy, 2017].

Clearly, as $z \to \infty$, the empirical distribution converges in probability to the true distribution and yet, implicit in the design of BLASTS is an assumption that a similar convergence statement holds for the plug-in estimator and the true rate-distortion function $\mathcal{R}(D)$. Fortunately, Harrison and Kontoyiannis [2008] have already proven general consistency results for the plug-in estimator of the rate-distortion function in two distinct settings of interest to sequential decision-making problems: (1) under a finite action set and (2) when the action set is a compact, separable metric space such that for each realization of the environment $\mathcal{E}$, $d(\mathcal{E}, \cdot)$ is continuous; naturally, the latter condition ensures consistency even under a continuous action space, although running the Blahut-Arimoto algorithm to optimize such a continuous channel output is non-trivial [Dauwels, 2005].

While the previous result is encouraging, it immediately begs the question of how many posterior samples $z$ are needed for the plug-in estimator $\widehat{\mathcal{R}}(D)$ to, with high probability, be an $\varepsilon$-accurate approximation of the true rate-distortion function $\mathcal{R}(D)$? While Palaiyanur and Sahai [2008] offer an answer to this question in their study of the uniform continuity of the rate-distortion function for discrete random variables, their analysis and corresponding sample-complexity bounds depend on a problem-specific parameter $\widetilde{D} = \min\limits_{(\mathcal{E},\chi):d(\mathcal{E},\chi)>0} d(\mathcal{E}, \chi)$ denoting the minimum non-zero distortion achieved by all environment-target pairs. While this constant does make sense in an information-theoretic context (for instance, under Hamming distortion, $\widetilde{D} = 1$), minimizing over all possible environments under a notion of distortion commensurate with expected squared regret engenders exceedingly small values of $\widetilde{D}$, rendering the associated bounds vacuous. To remedy this issue, we consider a minor modification to the statement of Lemma 2 by Palaiyanur and Sahai [2008], replacing their more general result that carries an unfavorable dependence on $\widetilde{D}$ with a less general result in terms of

$$D(\mathbb{P}_{\mathcal{E}}) = \min_{\mathcal{E}:\mathbb{P}_{\mathcal{E}}(\mathcal{E})>0} \min_{\chi:d(\mathcal{E},\chi)>0} d(\mathcal{E}, \chi)$$

When the distortion function is aligned with the regret between the optimal action and target action, $D(\mathbb{P}_\mathcal{E})$ corresponds to the worst-case action gap between the best and second-best actions [Farahmand, 2011, Agrawal and Goyal, 2013, Bellemare et al., 2016] based on the agent's current posterior over the environment. It is important to note that while our adjustment to Lemma 2 (and its corresponding effect on Lemma 5) of Palaiyanur and Sahai [2008] carries meaningful semantics for our sequential decision-making setting, the proof of the modified result still follows exactly as in their original paper[2], which leverages the uniform continuity of entropy.

**Lemma 3** (Lemma 2 - [Palaiyanur and Sahai, 2008]). *Let $\mathcal{E}, \chi$ be discrete random variables. Let $\mathbb{P}_\mathcal{E}$ denote the agent's current posterior over $\mathcal{E}$ and let $\widehat{\mathbb{P}}_\mathcal{E}$ be the empirical distribution. If $||\mathbb{P}_\mathcal{E} - \widehat{\mathbb{P}}_\mathcal{E}||_1 \leq \frac{D(\mathbb{P}_\mathcal{E})}{4}$, then for any $D \geq 0$,*

$$|\mathcal{R}(D) - \widehat{\mathcal{R}}(D)| \leq \frac{7}{D(\mathbb{P}_\mathcal{E})}||\mathbb{P}_\mathcal{E} - \widehat{\mathbb{P}}_\mathcal{E}||_1 \log\left(\frac{|\Sigma||\Xi|}{||\mathbb{P}_\mathcal{E} - \widehat{\mathbb{P}}_\mathcal{E}||_1}\right).$$

**Corollary 1** (Lemma 5 - [Palaiyanur and Sahai, 2008]). *For any $\delta \in (0, 1), \varepsilon \in (0, \log(\Sigma))$ let $\mathbb{P}_\mathcal{E} \in \Delta(\Sigma)$ be the current posterior over $\mathcal{E}$. Additionally, let $\phi^{-1}$ denote the inverse of the function $\phi : [0, \frac{1}{2}] \to \mathbb{R}$ where $\phi(t) = t \log\left(\frac{|\Sigma||\Xi|}{t}\right)$. If*

$$z \geq \frac{2}{\phi^{-1}\left(\frac{\varepsilon D(\mathbb{P}_\mathcal{E})}{7}\right)^2}\left(\log\left(\frac{1}{\delta}\right) + |\Sigma|\log(2)\right), \text{ then } \mathbb{P}(|\mathcal{R}(D) - \widehat{\mathcal{R}}(D)| \geq \varepsilon) \leq \delta.$$

While the results discussed so far provide insight into the Blahut-Arimoto algorithm itself, it is worth examining the extent to which learning targets computed via rate-distortion theory improve over a hand-crafted learning target. While such an improvement is certainly expected based on the theoretical analyses of [Russo and Van Roy, 2018b, Arumugam and Van Roy, 2021], it has not yet been demonstrated empirically. Due to space constraints, we defer a presentation of such empirical results that elucidate the value of computing learning targets via rate-distortion theory to the appendix.

## 3.2 Blahut-Arimoto Information-Directed Sampling

While BLASTS is successful in leveraging optimal learning targets, we consider improving upon the exploration technique used for collecting information about those targets through the algorithmic design principle of information-directed sampling [Russo and Van Roy, 2018a]. Information-directed sampling (IDS) is an abstract objective for sequential decision-making agents where, at each time period, an agent computes a policy based on the current history $H_t$ that minimizes the information ratio

$$\pi_t = \min_{\pi \in \Delta(\mathcal{A})} \frac{\mathbb{E}_\pi\left[\mathbb{E}_\mathcal{E}\left[\overline{R}(A^\star) - \overline{R}(A_t) \mid H_t\right]\right]^2}{\mathbb{I}(A^\star; A_t, O_{t+1}|H_t = H_t)}.$$

Here, $\overline{R}(\cdot)$ denotes the mean reward associated with an input action. In words, the information ratio of a particular policy $\pi \in \Delta(\mathcal{A})$ weighs the squared expected regret of action $A_t \sim \pi$ relative to the optimal action $A^\star$ against the expected information gain about $A^\star$ by taking $A_t$. Note that the information gain term $\mathbb{I}(A^\star; A_t, O_{t+1}|H_t = H_t)$ conditions upon the particular realization of the agent's history at time period $t$, $H_t$. Relating this quantity to the more traditional notion of conditional mutual information only requires integrating over the randomness in $H_t$: $\mathbb{E}\left[\mathbb{I}(A^\star; A_t, O_{t+1}|H_t = H_t)\right] = \mathbb{I}(A^\star; A_t, O_{t+1}|H_t)$. For our purposes, we consider a modified information ratio that leverages the output of the Blahut-Arimoto algorithm.

In particular, we define Blahut-Arimoto Information-Directed Sampling (BLAIDS) as a general recipe for efficient exploration with optimal learning targets wherein an agent first uses its current posterior as a source distribution, computing a learning target $\chi$ via the Blahut-Arimoto algorithm. Subsequently, the agent employs IDS to balance optimal behavior and information gain about this target by minimizing the following information ratio

$$\Psi_t(\pi) = \frac{\mathbb{E}_\pi\left[\Delta_t(a)\right]^2}{\mathbb{I}(\chi; A_t, O_{t+1}|H_t = H_t)} \qquad \Delta_t(a) = \mathbb{E}_{\mathcal{E}, \chi}\left[\overline{R}(\chi) - \overline{R}(a) \mid H_t\right],$$

---

[2]Under a standard assumption that rewards are bounded in $[0, 1]$ and $d(\mathcal{E}, \chi)$ is a function of regret, the $d^\star$ term that appears in their original bound is equal to 1.

where the expectation is with respect to the joint distribution computed by the Blahut-Arimoto algorithm. Observe that the distortion function used here is given by $d(\mathcal{E}, \chi \mid H_t) = d(\mathcal{E}, A_\chi \mid H_t) = \mathbb{E}[(\overline{R}(A^\star) - \overline{R}(A_\chi))^2 \mid \mathcal{E}, H_t]$ where, conditioned on a realization of the environment $\mathcal{E}$, we may compute $\overline{R}(\cdot)$ [Arumugam and Van Roy, 2021]. One perspective on BLAIDS is that IDS, with its default focus on $A^\star$, will spend time acquiring too much information relative to BLAIDS that will adaptively guide IDS towards uncovering the minimum number of bits needed to achieve a desired level of sub-optimality. Naturally, BLAIDS is a generalization of IDS since, as $\beta \to \infty$, the Blahut-Arimoto algorithm computes the optimal action $A^\star$ for each realization of the environment from the input source distribution. Just as with IDS, BLAIDS is a design principle whose abstract objective must be made explicit in a computationally-tractable manner in order to yield a corresponding algorithm. One standard choice for IDS is to leverage a lower bound of mutual information by the variance in rewards [Russo and Van Roy, 2018a, Nikolov et al., 2018, Dwaracherla et al., 2020, Lu et al., 2021]. This results in an upper bound on the per-period information ratio that is suitable for minimization:

$$\Psi_t(\pi) \leq \frac{\mathbb{E}_\pi\left[\Delta_t(A_t)\right]^2}{\mathbb{E}\pi\left[v_t(A_t)\right]} \qquad v_t(A_t) = \mathbb{V}\left[\mathbb{E}\left[\overline{R}(A_t)|\chi, H_t\right]|H_t\right].$$

To see how this upper bound lends itself to a computationally-tractable instantiation of IDS, observe that with $z$ posterior samples, the Blahut-Arimoto algorithm results in a distribution over learning targets conditioned on each posterior sample of the environment $p(\chi = \tilde{a}|\mathcal{E} = e)$. Denoting the induced marginal over learning targets as $q(\tilde{a})$, we have

$$v_t(A_t = a) = \sum_{\tilde{a} \in \mathcal{A}} q(\tilde{a}) \left(\frac{1}{z}\sum_e \frac{p(\tilde{a}|e)}{q(\tilde{a})}\mathbb{E}_t\left[\overline{R}(A_t)|\chi = a, \mathcal{E} = e\right] - \frac{1}{z}\sum_e \mathbb{E}_t\left[\overline{R}(A_t)|\mathcal{E} = e\right]\right)^2.$$

Using this lower bound, we recover variance-BLAIDS as a computationally-tractable analogue to variance-IDS. Once again, to see that variance-BLAIDS generalizes variance-IDS note that as $\beta \to \infty$, recovery of the optimal action by the Blahut-Arimoto algorithm implies that for any realization of the environment and any $\tilde{a} \in \mathcal{A}$, $p(\tilde{a}|\mathcal{E} = e) \in \{0, 1\}$, identifying if $\tilde{a}$ is optimal in that realization of the environment. Consequently, $\sum_{\mathcal{E}} p(\tilde{a}|\mathcal{E}) = |\mathcal{E}_{a^\star = \tilde{a}}|$ and $q(\tilde{a}) = z^{-1}|\mathcal{E}_{a^\star = \tilde{a}}|$, where $\mathcal{E}^\star_{a^\star = \tilde{a}}$ denotes a partition of the posterior samples such that $a^\star \in \mathcal{A}$ is optimal for all environments $e \in \mathcal{E}_{a^\star = \tilde{a}}$. In the next section, we assess the extent to which variance-BLAIDS improves upon standard variance-IDS through the use of optimal learning targets.

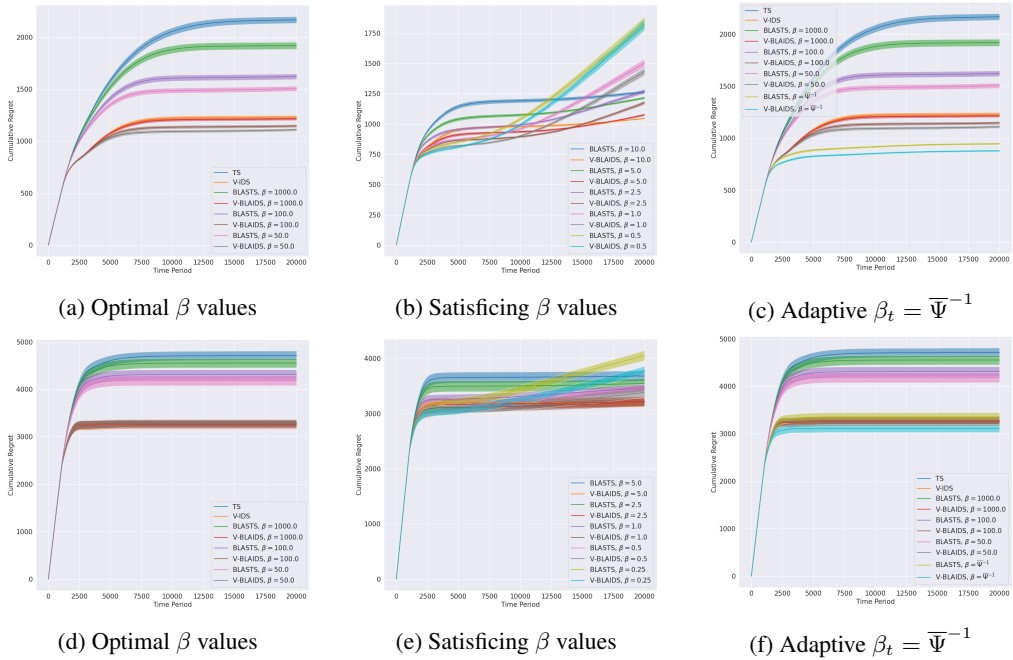

Figure 1: Regret curves for independent Bernoulli (top) and Gaussian (bottom) bandits with 50 arms.

## 4 Experiments

In our experimental setup, we examine independent Bernoulli and Gaussian bandit problems, sweeping over values of $\beta$ in order to examine the spectrum of policies induced by the various learning targets. Similar to BLASTS, BLAIDS is agnostic to the particular mechanism used for representing an agent's epistemic uncertainty. For our experiments, we found the use of a linear hypermodel [Dwaracherla et al., 2020] to be sufficient when optimized via Adam [Kingma and Ba, 2014] (learning rate of 0.001) with a noise variance of 0.1, prior variance of 1.0, and batch size of 1024. We adopt all other hyperparameters of [Arumugam and Van Roy, 2021] for running the Blahut-Arimoto algorithm [James et al., 2018][3] in each time period and run experiments via Google Colab with a GPU. In Figure 1, we report the cumulative regret in each time period across Thompson Sampling (TS), BLASTS, variance-IDS (V-IDS), and variance-BLAIDS (V-BLAIDS) where shading denotes 95% confidence intervals averaged over 10 random trials. For clarity, we split the figures in each problem between those values of $\beta$ which manage to achieve an optimal policy and those which learn a sub-optimal policy.

Our empirical results highlight opportunities for more efficient learning in light of information-directed exploration. Notably, the gap between BLASTS and variance-BLAIDS widens as $\beta$ increases, asserting more importance in pursuing a near-optimal learning target. Curiously, while the Bernoulli bandit does show small performance gains relative to variance-IDS when lowering $\beta$, such improvements do not appear in the Gaussian setting and present an interesting direction for future work to investigate the nature of information structures where learning an optimal policy can be improved through automated learning targets. That said, it is worth noting that the prior variance used in our experiments already matches the true variance of the environment and so, under a less-calibrated prior, there might be greater opportunity for variance-BLAIDS to shine over variance-IDS.

As in Arumugam and Van Roy [2021], we also entertain the idea of an intimate relationship between $\beta$ and the (best) information ratio $\overline{\Psi}_t = \min_{\pi \in \Delta(\mathcal{A})} \frac{\mathbb{E}_\pi[\Delta_t(a)]^2}{\mathbb{I}(A^\star; A_t, O_{t+1} | H_t = H_t)}$. We also find that the use of an adaptive $\beta_t = \overline{\Psi}_t^{-1}$ yields the best performance in both problem classes for learning an optimal policy. While the existence of a meaningful correspondence between $\beta$ and $\overline{\Psi}_t$ seems natural given their respective roles and aligned units (bits per squared unit of regret for the former and squared units of regret per bit for the latter), it is unclear if and when this scaling is correct. On the whole, these results serve as simple, fundamental sanity checks that confirm the importance of prudent information acquisition, even when in pursuit of a learning target besides the optimal action.

## 5 Conclusion

Learning targets are a general mechanism for guiding the information sought out by a sequential decision-making agent. As environments continue to grow in complexity so as to meet the demands of more challenging real-world tasks, so too will the need for learning targets that fall within the means of computationally-bounded agents. This work falls in with a new direction that leverages rate-distortion theory in order to forge such optimal learning targets and asks how such targets couple with existing methods for optimal information acquisition. Our work confirms that the computationally-tractable approaches for computing optimal learning targets are compatible with the well-studied principle of information-directed sampling. While we have found suitable answers for the multi-armed bandit setting, much remains to be understood about the nature of these learning targets themselves as well as how to extend these algorithms to handle target policies and other richer notions of learning objectives.

## Acknowledgments and Disclosure of Funding

Financial support from Army Research Office (ARO) grant W911NF2010055 is gratefully acknowledged.

---

[3] https://github.com/dit/dit

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
