# The Value of Information
# When Deciding What to Learn (Appendix)

**Dilip Arumugam**
Department of Computer Science
Stanford University
dilip@cs.stanford.edu

**Benjamin Van Roy**
Department of Electrical Engineering
Department of Management Science & Engineering
Stanford University
bvr@stanford.edu

## A  Related Work

This work primarily focuses on the design of sample-efficient sequential decision-making agents through principled methods of information acquisition and information representation [Lu et al., 2021]. While the use of information theory, and in particular rate-distortion theory, has been explored for the latter [Abel et al., 2019, Dong and Van Roy, 2018], this work aims to advance understanding of the role that rate-distortion theory plays in addressing the former challenge; we additionally hope that future work may further capitalize on these insights and techniques for information representation as well.

The challenge of efficient information acquisition is intimately tied to the exploration-exploitation trade-off that quintessentially appears in multi-armed bandit problems [Bubeck et al., 2012, Lattimore and Szepesvári, 2020]. Foremost among the various principled approaches for delicately negotiating this trade-off is the algorithmic design principle known as information-directed sampling (IDS) [Russo and Van Roy, 2018a]. IDS centers around a fundamental quantity known as the information ratio: the ratio in each time period of the squared expected regret and the expected information gain. The complimentary design principle naturally prescribed by this quantity is for an agent to, in each timestep, compute the policy that minimizes this information ratio. Accompanying the algorithmic simplicity that IDS espouses are the simple yet elegant theoretical analyses that it facilitates, which have been extended and adapted in numerous works [Lattimore and Szepesvári, 2019, Zimmert and Lattimore, 2019, Bubeck and Sellke, 2020, Lattimore and György, 2020, Kirschner et al., 2020]. While IDS has been extensively studied and the information ratio has been refined and generalized in numerous ways, there has been an almost exclusive focus on measuring information gained about the optimal action. While optimal actions are the prominent object of study throughout the sequential decision-making literature, the challenges of sample-efficient learning when faced with the complexity and scale of the real world motivate the need for fine-grained control over more generic notions of learning targets. This fact has been recognized by previous works [Russo and Van Roy, 2018b, Lu et al., 2021] which identify the utility of rate-distortion theory in overcoming these challenges, but offer no concrete algorithmic mechanisms for leveraging it fruitfully. Meanwhile, Arumugam and Van Roy [2021] succeed in addressing the algorithmic hurdle but only examine and analyze action selection via Thompson Sampling; our work closes this natural gap and offers a new generalization of IDS that accommodates effective information acquisition about alternative learning targets.

Finally, we note that the Blahut-Arimoto algorithm [Blahut, 1972, Arimoto, 1972] proves to be a critical algorithmic tool for enjoying the practical virtues of rate-distortion theory in sequential decision-making problems. The algorithm itself has been a popular object of study for its utility in the information-theory community as well as its efficacy as an alternating optimization algorithm [Boukris, 1973, Rose, 1994, Sayir, 2000, Matz and Duhamel, 2004, Niesen et al., 2007, Vontobel et al., 2008, Naja et al., 2009, Yu, 2010]. While we do not explore any extensions of the Blahut-Arimoto algorithm in this work due to their focus on improving computational efficiency,

35th Conference on Neural Information Processing Systems (NeurIPS 2021).

practitioners may find these computational advantages meaningful and potentially necessary when implementing and deploying BLAIDS for real-world applications.

# B  Background & Notation

In this section, we begin with an overview of several standard quantities in information theory as well as some useful facts. For more background on information theory, see Cover and Thomas [2012], Gray [2011]. All random variables are defined on a probability space $(\Omega, \mathbb{F}, \mathbb{P})$. For any random variable $X : \Omega \to \mathcal{X}$ taking values on a measurable space $(\mathcal{X}, \mathbb{X})$, we denote the associated (marginal) distribution of $X$ as

$$\mathbb{P}_X(A) = \mathbb{P}(X^{-1}(A)) = \mathbb{P}(\{\omega \mid X(\omega) \in A\}) \qquad \forall A \in \mathbb{X}.$$

We adopt analogous conventions for the joint and (regular) conditional probability measures, $\mathbb{P}_{X,Y}, \mathbb{P}_{X|Y}$, with respect to another random variable $Y$. For three random variables $X; Y;$ and $Z$, we define entropy; conditional entropy; mutual information; and conditional mutual information as follows:

$$\mathbb{H}(X) = -\mathbb{E}[\log(\mathbb{P}_X(\cdot))]$$
$$\mathbb{H}(Y|X) = -\mathbb{E}[\log(\mathbb{P}_{Y|X}(\cdot))]$$
$$\mathbb{I}(X; Y) = \mathbb{H}(X) - \mathbb{H}(X|Y) = \mathbb{H}(Y) - \mathbb{H}(Y|X)$$
$$\mathbb{I}(X; Y|Z) = \mathbb{H}(X|Z) - \mathbb{H}(X|Y, Z) = \mathbb{H}(Y|Z) - \mathbb{H}(Y|X, Z)$$

It is also useful to note another definition of mutual information through the Kullback-Leibler (KL) divergence:

$$\mathbb{I}(X; Y) = D_{\mathrm{KL}}(\mathbb{P}_{X,Y} \,||\, \mathbb{P}_X \times \mathbb{P}_Y) \qquad D_{\mathrm{KL}}(\mathbb{P} \,||\, \mathbb{Q}) = \begin{cases} \int \log\left(\frac{d\mathbb{P}}{d\mathbb{Q}}\right) d\mathbb{P} & \mathbb{P} \ll \mathbb{Q} \\ \infty & \mathbb{P} \not\ll \mathbb{Q} \end{cases},$$

where $\frac{d\mathbb{P}}{d\mathbb{Q}}$ denotes the Radon-Nikodym derivative of probability measure $\mathbb{P}$ with respect to $\mathbb{Q}$, with both measures defined on the same measurable space. Note that $\mathbb{P}_X \times \mathbb{P}_Y$ denotes the product measure over the associated marginals.

Consider a random variable $X : \Omega \to \mathcal{X}$ taking values on the measurable space $(\mathcal{X}, \mathbb{X})$ with an associated marginal distribution $\mathbb{P}_X$ that represents an information source. Similarly, define the random variable $\widehat{X} : \Omega \to \widehat{\mathcal{X}}$ that takes values on $(\widehat{\mathcal{X}}, \widehat{\mathbb{X}})$ and corresponds to a channel output. Given a known, measurable distortion measure $d : \mathcal{X} \times \widehat{\mathcal{X}} \mapsto \mathbb{R}_{\geq 0}$ and a desired upper bound on distortion $D$, the rate-distortion function is defined as:

$$\mathcal{R}(D) = \inf_{\mathbb{P}_{X,\widehat{X}} \in \Lambda} \mathbb{I}(X; \widehat{X}) \tag{1}$$

quantifying the minimum number of bits (on average) that must be communicated from $X$ across a channel in order to adhere to the specified expected distortion threshold $D$. Here, the infimum is taken over $\Lambda = \{\mathbb{P}_{X,\widehat{X}} \mid \forall A \in \mathbb{X} : \mathbb{P}_{X,\widehat{X}}(A \times \widehat{\mathcal{X}}) = \mathbb{P}_X(A) \text{ and } \mathbb{E}[d(X, \widehat{X})] \leq D\}$ representing the set of all joint distributions on the product space $(X \times \widehat{X}, \mathbb{X} \times \widehat{\mathbb{X}})$ whose corresponding marginal distribution on $X$ matches the original information source $\mathbb{P}_X$ while also satisfying the constraint on bounded expected distortion. Intuitively, a higher rate corresponds to requiring more bits of information and smaller information loss between $X$ and $\widehat{X}$, enabling higher-fidelity reconstruction (lower distortion); conversely, lower rates reflect more substantial information loss, potentially exceeding the tolerance on distortion $D$.

We will make use of the following facts:

**Fact 1** (Chain Rule for Radon-Nikodym Derivatives - Lemma 6.6 [Gray, 2009]). *Let $\mu, \nu, \rho$ be $\sigma$-finite probability measures on $(\Omega, \mathbb{F})$. If $\nu \ll \mu$ and $\mu \ll \rho$ with corresponding Radon-Nikodym derivatives $\frac{d\nu}{d\mu}, \frac{d\mu}{d\rho} < \infty$, then $\nu \ll \rho$ and has an associated Radon-Nikodym derivative $\rho$-a.s.*

$$\frac{d\nu}{d\rho} = \frac{d\nu}{d\mu}\frac{d\mu}{d\rho}.$$

**Fact 2** (Gibb's Inequality - [Cover and Thomas, 2012]). *For any two probability measures $\mathbb{P}, \mathbb{Q}$ on $(\Omega, \mathbb{F})$,*

$$D_{\mathrm{KL}}(\mathbb{P} \,||\, \mathbb{Q}) \geq 0.$$

**Fact 3** (Mutual Information as an Infimum over Product Measures - Corollary 7.12 [Gray, 2011]). *Let $\mathbb{P}_{X,Y}$ be a joint probability measure and let $\mathcal{M}$ denote the collection of all product measures $M_X \times M_Y$. Then*

$$\mathbb{I}(X;Y) = \inf_{M_X \times M_Y \in \mathcal{M}} D_{\mathrm{KL}}(\mathbb{P}_{X,Y} \,||\, M_X \times M_Y).$$

**Fact 4** (Uniform Continuity of Entropy - [Cover and Thomas, 2012]). *Let $\mathbb{P}, \mathbb{Q}$ be two discrete probability distributions (that is, probability mass functions) on the same measurable space $(\mathcal{X}, \mathbb{X})$. If*

$$||\mathbb{P} - \mathbb{Q}||_1 \leq \frac{1}{2},$$

*then*

$$|\mathbb{H}(\mathbb{P}) - \mathbb{H}(\mathbb{Q})| \leq ||\mathbb{P} - \mathbb{Q}||_1 \log\left(\frac{|\mathcal{X}|}{||\mathbb{P} - \mathbb{Q}||_1}\right).$$

## C  Technical Proofs

**Lemma 1** (Lemma 1.2 [Csiszár, 1974]).

$$\mathcal{R}(D) = \max_{\beta \geq 0} \mathcal{F}(\beta) - \beta D$$

*The value of $\beta$ that achieves this maximum is characterized as being associated with the distortion threshold $D$. Conversely, a joint distribution $\mathbb{P}_{\mathcal{E}, \chi}$ that achieves the infimum of $\mathcal{F}(\beta)$ has an associate rate $\mathcal{R}(D) = \mathbb{I}(\mathbb{P}_{\mathcal{E}, \chi})$ where $\beta$ is associated with $D = \mathbb{E}_{\mathbb{P}_{\mathcal{E}, \chi}}[d(\mathcal{E}, \chi)] \triangleq \int d(\mathcal{E}, \chi) d\mathbb{P}_{\mathcal{E}, \chi}(\mathcal{E}, \chi)$.*

*Proof.* The correspondence between values of $\beta$ and values of $D$ is a geometric argument. We have that for any fixed $D \geq 0$ and for all $\beta \geq 0$,

$$
\begin{aligned}
\mathcal{R}(D) &= \inf_{\mathbb{P}_{\mathcal{E}, \chi} \in \Lambda : \mathbb{E}[d(\mathcal{E}, \chi)] \leq D} \mathbb{I}(\mathbb{P}_{\mathcal{E}, \chi}) \\
&= \inf_{\mathbb{P}_{\mathcal{E}, \chi} \in \Lambda : \mathbb{E}[d(\mathcal{E}, \chi)] \leq D} \left( \mathbb{I}(\mathbb{P}_{\mathcal{E}, \chi}) + \beta \mathbb{E}_{\mathbb{P}_{\mathcal{E}, \chi}}[d(\mathcal{E}, \chi)] - \beta \mathbb{E}_{\mathbb{P}_{\mathcal{E}, \chi}}[d(\mathcal{E}, \chi)] \right) \\
&\geq \inf_{\mathbb{P}_{\mathcal{E}, \chi} \in \Lambda : \mathbb{E}[d(\mathcal{E}, \chi)] \leq D} \left( \mathbb{I}(\mathbb{P}_{\mathcal{E}, \chi}) + \beta \mathbb{E}_{\mathbb{P}_{\mathcal{E}, \chi}}[d(\mathcal{E}, \chi)] \right) - \beta D \\
&\geq \inf_{\mathbb{P}_{\mathcal{E}, \chi} \in \Lambda} \left( \mathbb{I}(\mathbb{P}_{\mathcal{E}, \chi}) + \beta \mathbb{E}_{\mathbb{P}_{\mathcal{E}, \chi}}[d(\mathcal{E}, \chi)] \right) - \beta D \\
&= \mathcal{F}(\beta) - \beta D
\end{aligned}
$$

This inequality coupled with the fact that $\mathcal{R}(D)$ is convex implies the existence of a straight line with slope $\beta_D$ passing through the point $(D, \mathcal{R}(D))$ such that $\mathcal{F}(\beta) - \beta_D D = \mathcal{R}(D)$ with no points above the rate-distortion curve. Consequently, the notion of a particular $\beta_D$ associated with $D$ implies that for all $D'$,

$$\mathcal{R}(D') + \beta_D D' \geq \mathcal{R}(D) + \beta_D D.$$

To see that this lower bound is achieved by the $\beta_D$ value associated with a given threshold $D$, let $\mathbb{P}_{\mathcal{E}, \chi}$ be a joint distribution with expected distortion $D'$ such that for $\varepsilon > 0$,

$$\mathbb{I}(\mathbb{P}_{\mathcal{E}, \chi}) + \beta_D \mathbb{E}_{\mathcal{E}, \chi}[d(\mathcal{E}, \chi)] \leq \mathcal{F}(\beta_D) + \varepsilon.$$

Using the previous two inequalities, we have

$$\mathcal{F}(\beta_D) + \varepsilon \geq \mathcal{R}(D') + \beta_D D' \geq \mathcal{R}(D) + \beta_D D \geq \mathcal{F}(\beta_D)$$

Since our choice of $\varepsilon$ was arbitrary, taking the limit as $\varepsilon \to 0$ yields

$$\mathcal{F}(\beta_D) = \mathcal{R}(D) + \beta_D D.$$

Notably, the Blahut-Arimoto algorithm for computing rate-distortion functions operates with a specification of $\beta$, rather than $D$. Let $\mathbb{P}^\star_{\mathcal{E},\chi}$ be the joint distribution such that

$$\mathcal{F}(\beta) = \inf_{\mathbb{P}_{\mathcal{E},\chi} \in \Lambda} \left( \mathbb{I}(\mathbb{P}_{\mathcal{E},\chi}) + \beta \mathbb{E}_{\mathbb{P}_{\mathcal{E},\chi}}[d(\mathcal{E},\chi)] \right) = \mathbb{I}(\mathbb{P}^\star_{\mathcal{E},\chi}) + \beta \mathbb{E}_{\mathbb{P}^\star_{\mathcal{E},\chi}}[d(\mathcal{E},\chi)].$$

Defining $D_\beta = \mathbb{E}_{\mathbb{P}^\star_{\mathcal{E},\chi}}[d(\mathcal{E},\chi)]$ and $R_\beta = \mathbb{I}(\mathbb{P}^\star_{\mathcal{E},\chi})$, we have the lower bound

$$\mathcal{F}(\beta) = R_\beta + \beta D_\beta \geq \inf_{\mathbb{P}_{\mathcal{E},\chi} \in \Lambda: \mathbb{E}[d(\mathcal{E},\chi)] \leq D_\beta} \mathbb{I}(\mathbb{P}_{\mathcal{E},\chi}) + \beta D_\beta = \mathcal{R}(D_\beta) + \beta D_\beta.$$

For the upper bound, we have

$$\begin{aligned}
\mathcal{F}(\beta) &= \inf_{\mathbb{P}_{\mathcal{E},\chi} \in \Lambda} \left( \mathbb{I}(\mathbb{P}_{\mathcal{E},\chi}) + \beta \mathbb{E}_{\mathbb{P}_{\mathcal{E},\chi}}[d(\mathcal{E},\chi)] \right) \\
&= \inf_{\mathbb{P}_{\mathcal{E},\chi} \in \Lambda: \mathbb{E}[d(\mathcal{E},\chi)] \leq D_\beta} \left( \mathbb{I}(\mathbb{P}_{\mathcal{E},\chi}) + \beta \mathbb{E}_{\mathbb{P}_{\mathcal{E},\chi}}[d(\mathcal{E},\chi)] \right) \\
&\leq \inf_{\mathbb{P}_{\mathcal{E},\chi} \in \Lambda: \mathbb{E}[d(\mathcal{E},\chi)] \leq D_\beta} \left( \mathbb{I}(\mathbb{P}_{\mathcal{E},\chi}) + \beta D_\beta \right) \\
&= \inf_{\mathbb{P}_{\mathcal{E},\chi} \in \Lambda: \mathbb{E}[d(\mathcal{E},\chi)] \leq D_\beta} \mathbb{I}(\mathbb{P}_{\mathcal{E},\chi}) + \beta D_\beta \\
&= \mathcal{R}(D_\beta) + \beta D_\beta
\end{aligned}$$

Putting both inequalities together shows that $\mathcal{F}(\beta) = \mathcal{R}(D_\beta) + \beta D_\beta$, which implies that $\beta$ is associated with $D_\beta$ and $\mathcal{R}(D_\beta) = \mathcal{F}(\beta) - \beta D_\beta$. $\qquad\square$

**Lemma 2** (Lemma 1.2 [Csiszár, 1974]). *Let $\mathbb{Q}_\chi$ be an arbitrary marginal distribution over $\chi$. Moreover, let $\mathbb{P}_{\mathcal{E},\chi}$ be an arbitrary joint distribution over $\mathcal{E}, \chi$ with an associated marginal distribution $\mathbb{P}_\chi$ and take $\mathbb{Q}_{\mathcal{E},\chi}$ to be the joint distribution as defined in Equation* **??**. *Then,*

$$\begin{aligned}
\mathcal{J}(\mathbb{P}_{\mathcal{E},\chi}, \mathbb{Q}_\chi, \beta) &\geq \mathcal{J}(\mathbb{P}_{\mathcal{E},\chi}, \mathbb{P}_\chi, \beta) \\
\mathcal{J}(\mathbb{P}_{\mathcal{E},\chi}, \mathbb{Q}_\chi, \beta) &\geq \mathcal{J}(\mathbb{Q}_{\mathcal{E},\chi}, \mathbb{Q}_\chi, \beta)
\end{aligned}$$

*Proof.* For the first inequality, recall that $\mathbb{P}_\chi$ is the true marginal distribution over learning targets induced by $\mathbb{P}_{\mathcal{E},\chi}$. Moreover, by definition, we have that

$$\mathcal{J}(\mathbb{P}_{\mathcal{E},\chi}, \mathbb{P}_\chi, \beta) = D_{\mathrm{KL}}(\mathbb{P}_{\mathcal{E},\chi} || \mathbb{P}_{\mathcal{E}} \times \mathbb{P}_\chi) + \beta \mathbb{E}_{\mathbb{P}_{\mathcal{E},\chi}}[d(\mathcal{E},\chi)] = \mathbb{I}(\mathcal{E};\chi) + \beta \mathbb{E}_{\mathbb{P}_{\mathcal{E},\chi}}[d(\mathcal{E},\chi)].$$

Therefore, by Fact 3, it follows that

$$\mathcal{J}(\mathbb{P}_{\mathcal{E},\chi}, \mathbb{Q}_\chi, \beta) = \mathcal{J}(\mathbb{P}_{\mathcal{E},\chi}, \mathbb{P}_\chi, \beta) + D_{\mathrm{KL}}(\mathbb{P}_\chi || \mathbb{Q}_\chi).$$

The inequality then follows immediately by Fact 2.

For the second inequality, we have

$$\begin{aligned}
\mathcal{J}(\mathbb{P}_{\mathcal{E},\chi}, \mathbb{Q}_\chi, \beta) &= D_{\mathrm{KL}}(\mathbb{P}_{\mathcal{E},\chi} || \mathbb{P}_{\mathcal{E}} \times \mathbb{Q}_\chi) + \beta \mathbb{E}_{\mathbb{P}_{\mathcal{E},\chi}}[d(\mathcal{E},\chi)] \\
&= \int \log\left( \frac{d\mathbb{P}_{\mathcal{E},\chi}}{d\mathbb{P}_{\mathcal{E}} \times \mathbb{Q}_\chi}(\mathcal{E},\chi) \right) d\mathbb{P}_{\mathcal{E},\chi}(\mathcal{E},\chi) - \int \log\left( \exp\left(-\beta d(\mathcal{E},\chi)\right) \right) d\mathbb{P}_{\mathcal{E},\chi}(\mathcal{E},\chi) \\
&= \int \log\left( \frac{d\mathbb{P}_{\mathcal{E},\chi}}{d\mathbb{P}_{\mathcal{E}} \times \mathbb{Q}_\chi}(\mathcal{E},\chi) \right) d\mathbb{P}_{\mathcal{E},\chi}(\mathcal{E},\chi) - \int \log\left( \alpha_{\mathbb{Q}_\chi,\beta}(\mathcal{E}) \exp\left(-\beta d(\mathcal{E},\chi)\right) \right) d\mathbb{P}_{\mathcal{E},\chi}(\mathcal{E},\chi) + \int \log\left( \alpha_{\mathbb{Q}_\chi,\beta}(\mathcal{E}) \right) d\mathbb{P}_{\mathcal{E}}(\mathcal{E}) \\
&= \int \log\left( \frac{d\mathbb{P}_{\mathcal{E},\chi}}{d\mathbb{P}_{\mathcal{E}} \times \mathbb{Q}_\chi}(\mathcal{E},\chi) \right) d\mathbb{P}_{\mathcal{E},\chi}(\mathcal{E},\chi) - \int \log\left( \frac{d\mathbb{Q}_{\mathcal{E},\chi}}{d\mathbb{P}_{\mathcal{E}} \times \mathbb{Q}_\chi}(\mathcal{E},\chi) \right) d\mathbb{P}_{\mathcal{E},\chi}(\mathcal{E},\chi) + \int \log\left( \alpha_{\mathbb{Q}_\chi,\beta}(\mathcal{E}) \right) d\mathbb{P}_{\mathcal{E}}(\mathcal{E}) \\
&= \int \log\left( \frac{d\mathbb{P}_{\mathcal{E},\chi}}{d\mathbb{P}_{\mathcal{E}} \times \mathbb{Q}_\chi}(\mathcal{E},\chi) \cdot \frac{d\mathbb{P}_{\mathcal{E}} \times \mathbb{Q}_\chi}{d\mathbb{Q}_{\mathcal{E},\chi}}(\mathcal{E},\chi) \right) d\mathbb{P}_{\mathcal{E},\chi}(\mathcal{E},\chi) + \int \log\left( \alpha_{\mathbb{Q}_\chi,\beta}(\mathcal{E}) \right) d\mathbb{P}_{\mathcal{E}}(\mathcal{E}) \\
&= \int \log\left( \frac{d\mathbb{P}_{\mathcal{E},\chi}}{d\mathbb{Q}_{\mathcal{E},\chi}}(\mathcal{E},\chi) \right) d\mathbb{P}_{\mathcal{E},\chi}(\mathcal{E},\chi) + \int \log\left( \alpha_{\mathbb{Q}_\chi,\beta}(\mathcal{E}) \right) d\mathbb{P}_{\mathcal{E}}(\mathcal{E}) \\
&= D_{\mathrm{KL}}(\mathbb{P}_{\mathcal{E},\chi} || \mathbb{Q}_{\mathcal{E},\chi}) + \int \log\left( \alpha_{\mathbb{Q}_\chi,\beta}(\mathcal{E}) \right) d\mathbb{P}_{\mathcal{E}}(\mathcal{E})
\end{aligned}$$

where the sixth line employs the chain rule for Radon-Nikodym derivatives (Fact 1). Expanding the second term and using the fact marginal distribution over $\mathcal{E}$ induced by $\mathbb{Q}_{\mathcal{E},\chi}$ is the same as $\mathbb{P}_\mathcal{E}$, we have

$$
\begin{aligned}
\int \log\left(\alpha_{\mathbb{Q}_\chi,\beta}(\mathcal{E})\right) d\mathbb{P}_\mathcal{E}(\mathcal{E}) &= \int \log\left(\alpha_{\mathbb{Q}_\chi,\beta}(\mathcal{E})\right) d\mathbb{Q}_{\mathcal{E},\chi}(\mathcal{E},\chi) \\
&= \int \log\left(\alpha_{\mathbb{Q}_\chi,\beta}(\mathcal{E})\exp\left(-\beta d(\mathcal{E},\chi)\right)\exp\left(\beta d(\mathcal{E},\chi)\right)\right) d\mathbb{Q}_{\mathcal{E},\chi}(\mathcal{E},\chi) \\
&= \int \log\left(\alpha_{\mathbb{Q}_\chi,\beta}(\mathcal{E})\exp\left(-\beta d(\mathcal{E},\chi)\right)\right) d\mathbb{Q}_{\mathcal{E},\chi}(\mathcal{E},\chi) + \int \log\left(\exp\left(\beta d(\mathcal{E},\chi)\right)\right) d\mathbb{Q}_{\mathcal{E},\chi}(\mathcal{E},\chi) \\
&= \int \log\left(\frac{d\mathbb{Q}_{\mathcal{E},\chi}}{d\mathbb{P}_\mathcal{E}\times\mathbb{Q}_\chi}(\mathcal{E},\chi)\right) d\mathbb{Q}_{\mathcal{E},\chi}(\mathcal{E},\chi) + \beta\int d(\mathcal{E},\chi) d\mathbb{Q}_{\mathcal{E},\chi}(\mathcal{E},\chi) \\
&= D_{\mathrm{KL}}(\mathbb{Q}_{\mathcal{E},\chi}\,||\,\mathbb{P}_\mathcal{E}\times\mathbb{Q}_\chi) + \beta\mathbb{E}_{\mathbb{Q}_{\mathcal{E},\chi}}\left[d(\mathcal{E},\chi)\right] \\
&= \mathcal{J}(\mathbb{Q}_{\mathcal{E},\chi},\mathbb{Q}_\chi,\beta)
\end{aligned}
$$

Substituting back and applying Fact 2 once more yields the inequality

$$
\begin{aligned}
\mathcal{J}(\mathbb{P}_{\mathcal{E},\chi},\mathbb{Q}_\chi,\beta) &= D_{\mathrm{KL}}(\mathbb{P}_{\mathcal{E},\chi}\,||\,\mathbb{Q}_{\mathcal{E},\chi}) + \int \log\left(\alpha_{\mathbb{Q}_\chi,\beta}(\mathcal{E})\right) d\mathbb{P}_\mathcal{E}(\mathcal{E}) \\
&= D_{\mathrm{KL}}(\mathbb{P}_{\mathcal{E},\chi}\,||\,\mathbb{Q}_{\mathcal{E},\chi}) + \mathcal{J}(\mathbb{Q}_{\mathcal{E},\chi},\mathbb{Q}_\chi,\beta) \\
&\geq \mathcal{J}(\mathbb{Q}_{\mathcal{E},\chi},\mathbb{Q}_\chi,\beta)
\end{aligned}
$$

$\square$

**Lemma 3** (Lemma 2 - [Palaiyanur and Sahai, 2008]). *Let $\mathcal{E},\chi$ be discrete random variables. Let $\mathbb{P}_\mathcal{E}$ denote the agent's current posterior over $\mathcal{E}$ and let $\widehat{\mathbb{P}}_\mathcal{E}$ be the empirical distribution. If $||\mathbb{P}_\mathcal{E}-\widehat{\mathbb{P}}_\mathcal{E}||_1 \leq \frac{D(\mathbb{P}_\mathcal{E})}{4}$, then for any $D\geq 0$,*

$$
|\mathcal{R}(D)-\widehat{\mathcal{R}}(D)| \leq \frac{7}{D(\mathbb{P}_\mathcal{E})}||\mathbb{P}-\widehat{\mathbb{P}}_\mathcal{E}||_1 \log\left(\frac{|\Sigma||\Xi|}{||\mathbb{P}_\mathcal{E}-\widehat{\mathbb{P}}_\mathcal{E}||_1}\right).
$$

*Proof.* The proof of this augmented result still follows the same argument outlined in Section V of Palaiyanur and Sahai [2008]. $\square$

**Corollary 1** (Lemma 5 - [Palaiyanur and Sahai, 2008]). *For any $\delta\in(0,1),\varepsilon\in(0,\log(\Sigma))$ let $\mathbb{P}_\mathcal{E}\in\Delta(\Sigma)$ be the current posterior over $\mathcal{E}$. Additionally, let $\phi^{-1}$ denote the inverse of the function $\phi:[0,\frac{1}{2}]\to\mathbb{R}$ where $\phi(t)=t\log\left(\frac{|\Sigma||\Xi|}{t}\right)$. If*

$$
z \geq \frac{2}{\phi^{-1}\left(\frac{\varepsilon D(\mathbb{P}_\mathcal{E})}{7}\right)^2}\left(\log\left(\frac{1}{\delta}\right)+|\Sigma|\log(2)\right),
$$

*then*

$$
\mathbb{P}(|\mathcal{R}(D)-\widehat{\mathcal{R}}(D)|\geq\varepsilon)\leq\delta.
$$

*Proof.*

$$
\mathbb{P}(|\mathcal{R}(D)-\widehat{\mathcal{R}}(D)|\geq\varepsilon) \leq \mathbb{P}\left(||\mathbb{P}_\mathcal{E}-\widehat{\mathbb{P}}_\mathcal{E}||_1\geq\phi^{-1}\left(\frac{\varepsilon D(\mathbb{P}_\mathcal{E})}{7}\right)\right) \leq 2^{|\Sigma|}\exp\left(-\frac{z}{2}\phi^{-1}\left(\frac{\varepsilon D(\mathbb{P}_\mathcal{E})}{7}\right)\right)^2,
$$

where the first inequality follows from Lemma 3 and the second inequality follows as Theorem 2.1 of [Weissman et al., 2003]. Setting the left-hand side equal to $\delta$ and re-arranging terms yields the inequality. $\square$

## D   On the Optimality of Rate-Distortion-Theoretic Learning Targets

We examine independent Bernoulli and Gaussian bandits with 50 arms and sweep across numerous $\beta$ values to trace the shape of the resulting rate-distortion curves associated with the various learning targets induced at the first time period. We compare this to the hand-crafted target action of Russo and Van Roy [2018b], Lu et al. [2021] that simply takes the first action whose average reward $\overline{R}_a$ is within $\varepsilon \geq 0$ of the optimal $\overline{R}^\star$:

$$\chi = \min\{a \in \mathcal{A} \mid \overline{R}_a \geq \overline{R}^\star - \varepsilon\}.$$

Analogously sweeping over values of $\varepsilon$ generates the results of Figure 1 where we notice a substantially improved information-performance trade-off from the Blahut-Arimoto algorithm. Such results highlight the importance of rate-distortion theory in yielding optimized learning targets that may otherwise be difficult for agent designers to engineer by hand.

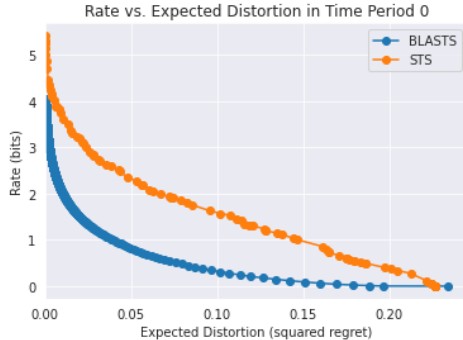
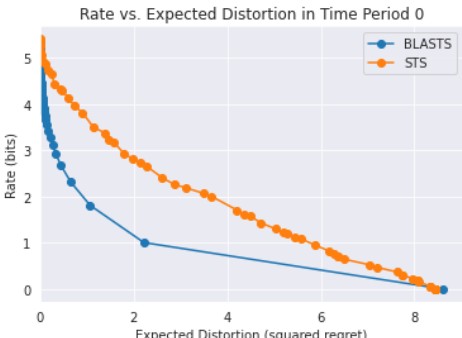

(a) Independent Bernoulli bandit with 50 arms.   (b) Independent Gaussian bandit with 50 arms.

Figure 1: Comparing target actions computed via the Blahut-Arimoto actions (BLASTS) vs. the hand-crafted target action (STS) examined by Russo and Van Roy [2018b], Lu et al. [2021].