# OpenReview forum: "The Value of Information When Deciding What to Learn"
_NeurIPS.cc/2021/Conference — NeurIPS 2021 Poster_

### Official Review · Reviewer_z9LJ · 2021-07-08

**Rating:** 6
**Confidence:** 2

**Summary:**

This papers studies the fundamental problem in sequential decision making problems related to exploration. Previous work have used Thomson sampling in these cases however, they are suboptimal.
This work uses rate distortion theory and in generalizes previous Blahut-Aritomoto and Information-directed sampling  algorithms with a novel BLAIDS approach.
I have some suggestions in terms of presentation to improve readability and clarity.

**Limitations And Societal Impact:**

No discussion on the topic

**Main Review:**

Originality
- Highly relevant problem in decision making

Clarity
- I had a hard time following the paper because in not familiar withe the background.
I would have like to see a definition of action-gap.
I would have liked to know more details on IDS.

- I also think the structure of the paper is different from what I'm used to, for example, there are key statements are kind of "hidden" in the paper (in between sections and paragraphs).

"We leave the  question of how to adaptively compute optimal target policies in a computationally-tractable manner to future work and, instead, turn our attention to the multi-armed bandit setting where we may more feasibly study the coupling of optimal learning targets with optimal information acquisition."

"This procedure is precisely the classic Blahut-Arimoto algorithm [Blahut, 1972,Arimoto, 1972], generalized beyond the assumptions of discrete source and channel output random variables."

"We define Blahut-Arimoto Information-Directed Sampling (BLAIDS) as a general recipe for efficient exploration with optimal learning targets wherein an agent first uses its current posterior as a source distribution, computing a learning target χ via the Blahut-Arimoto algorithm."

-Maybe you could move Section 2.4 at the end of the paper as Disscussion


Significance
- The results are presented on multi-armed bandits (MAB) the RL case is not studied in detail.
- The paper takes theoretical results already proven to the MAB case.


**Time Spent Reviewing:**

4

---

### Official Review · Reviewer_pEia · 2021-07-16

**Rating:** 6
**Confidence:** 4

**Summary:**

**Update after author response and discussion**: I am happy to see most of my assumptions positively confirmed and clarified by the authors, as well as the criticism by other reviewers being sufficiently addressed. Some of my improvements remain valid, but I do not consider them crucial for publication. I therefore stick with my original score but raise my confidence - I am in favor of accepting the paper.

**High-level summary**

The paper addresses the question of learning to act well in finite-horizon sequential tasks in unknown environments under computational limitations. This requires solving two problems. (1) In principle, full knowledge about the environment allows agents to act optimally. However, actions that lead to large information gain about the environment often come at the expense of incurring low immediate rewards - trading off these two considerations is non-trivial and has been formalized previously via information directed sampling (IDS), which aims to find policies that minimize the ratio between squared expected regret (of the policy compared to the optimal policy) and information gain (ideally leading to small regret and large information gain). (2) Reducing all (epistemic) uncertainty about the environment can incur a considerable amount of exploratory actions, particularly in complex environments. Often, a less informative (lossy) compression of the environment allows for learning policies that perform very well with considerably lower “exploration cost” (which is particularly relevant for computationally bounded agents). Such a compression has been previously introduced under the name ‘learning target’ which is computed via a (stochastic) mapping from the environment to the learning target random variable. A good learning target requires considerably less information (fewer bits) to be extracted from the environment (compared to reducing all uncertainty about the full environment) while at the same time ensuring that policies trained to maximize the learning target achieve good returns in the actual environments. This trade-off between low reduction in performance and high reduction in sample complexity of exploration has recently been formalized via a rate distortion theory, where the distortion is the expected regret between the policy trained on the learning target and the optimal policy (given an observation history), and the rate is the information that needs to be extracted from the environment to reduce uncertainty about the learning target variable sufficiently.

**Concrete summary**

Previous work (BLASTS) derived an improved Thompson sampling scheme based on learning targets obtained by solving the rate-distortion trade-off via Blahut-Arimoto iterations. While BLASTS provides a principled solution to extract good learning targets, the resulting policies are understood to perform poorly in terms of exploration (increasing informativity of the learning target). The current paper improves upon this by combining IDS (which explores well by taking into account potential information gain and overall performance improvements) with automatically learned targets via Blahut-Arimoto iterations (which solve the rate-distortion trade-off). This leads to a method with a hyper-parameter that allows the designer to put more or less emphasis on keeping the required information acquisition sample complexity low (thus potentially incorporating a range of computational budgets) - the latter of course comes at the cost of potential reductions in performance (returns) of the learned policy compared to an optimal policy (in terms of cumulative regret over the course of learning, but potentially also for the converged solutions if the computational budget is too low). Results are shown for two 50-armed bandits (Bernoulli and Gaussian) where the method compares favorably with other methods (including BLASTS), and on par with IDS on Gaussian bandits.

**Contributions:**

1) Combination of IDS with Blahut-Arimoto learning targets. Both parts are theoretically well motivated and the combination makes sense and leads to a principled theoretical method. To obtain a practical implementation an approximation is needed - the approximation is sensible, though perhaps a bit coarse, but it is theoretically well backed up). Significance: results on the Bernoulli bandit look good, but performance gains compared to IDS don’t carry over to the Gaussian case. Since it remains unclear why, the potential impact remains limited (does the method sound good in theory but easily runs into practical problems, or is there a good reason why we would not expect to see a difference in the Gaussian case?).

2) Solid theoretical foundations - given a constraint on the distortion (corresponding to a value of the hyper-parameter \beta) there is a theoretical guarantee that the learning target is optimal in the sense that there exists no other learning signal target that achieves the same distortion but requires lower rate (“bits extracted from the environment”) - or dually, given a computational budget (set by inverse \beta) no other learning target can lead to higher returns. Significance: given the limited empirical results it remains somewhat unclear to which degree the theoretical results carry over in practice - it cannot be ruled out that (slightly) sub-optimal learning targets could lead to better results in particular cases (e.g. leading to more stable learning, easier to find corresponding policies, etc.). This remains to be seen in the future, but having a solid theoretical background is important - even heuristics might benefit from the theoretical insight presented here.

3) Review of (continuous) rate-distortion theory and the necessary background for the method for a general ML/RL audience (non info-theory experts). Significance: important contribution in itself, though the material is still a bit dense and might need further unpacking / illustrations of the key intuitions for e.g. a general RL practitioner audience. Also, the latter target audience might not be convinced to invest time since the method is currently limited to (non-contextual) bandits.

**Ethical Concerns:**

No ethical concerns.

**Limitations And Societal Impact:**

Some limitations are pointed out throughout the text. It would not hurt to have a paragraph at the end of the paper dedicated to current limitations and shortcomings (in terms of the concretely implemented method, including e.g. the mutual info estimation). Societal impact: not applicable.

**Main Review:**

**Originality, Quality, Clarity, Correctness:**

The paper is well written, though some parts are technically dense (but I don’t see an easy solution to this given the page limitations). Importantly, the paper does a great job of explaining the necessary background (with a great section summarizing continuous rate distortion theory in a way that makes the rest of the manuscript much more accessible), the fundamental problems, and previous solutions to these individual problems. To the best of my knowledge, the proposed method is novel - it mainly consists of combining a recent idea BLASTS with the more widely known IDS. This is clearly pointed out in the text and the combination is sensible, and compared against both previous methods. I have not checked the proofs in detail, but did not spot a result in the main paper that raised my suspicions.

**Verdict:**

Overall the paper addresses a timely and important problem, and provides a very principled solution approach, which I have greatly enjoyed. The main innovation in the paper is the combination of two existing methods, which is somewhat incremental. The paper compensates for this by clearly introducing the problem and providing the necessary background for a broader audience to understand the solution without having to resort to other publications. My main criticism is that results in the Gaussian case are on par, but not better than IDS. It would be very good to know whether that is due to some fundamental problem (from translating the theoretical principles into a practical algorithm), or whether this is somewhat expected for the Gaussian case. Overall I think the material is ready for publication, but given the incremental nature might receive most attention in a smaller sub-community (though I highly appreciate the attempt to make the material accessible to a wider audience). I would like to see a unified manuscript (including IDS, parts of the orig. learning targets paper, and BLASTS) written with the same target audience in mind, and more space for intuitive illustrations of the core concepts - but this would obviously not fit into a conference-format paper.

**Pros:**

 * Clearly defined problem, well described partial solutions and their shortcomings, sensible and well explained main innovation.
 * Extensive background section, akin to a mini-review on continuous rate-distortion theory - which is crucial to make the paper accessible to a wider audience.
 * Empirical results show some support of the theoretical claims - the main method seems to work and perform better than previous methods (except for one case).

**Cons:**

 * Somewhat incremental approach (given IDS and BLASTS) - but I think there is sufficient novelty to warrant publication (since there are also clear shortcomings of both predecessor methods that are addressed).
 * Applications limited to (non-contextual) bandits for now - extensions to more complex RL problems require major technical innovations/approximations (but this is also clearly discussed in the paper)
 * Empirical results are limited to two simple bandits
 * Comparison with IDS on Gaussian bandits somewhat inconclusive

**Improvements:**

1) It would be good to investigate the reasons for the results compared to IDS on the Gaussian bandit. This could be either a theoretical investigation (are there any reasons why we might expect these results) or an empirical investigation (do other bandit settings lead to similar results, or is this an observation limited to the Gaussian case?). Ideally both.

2) Though space is already tight, it would be nice to show results for more bandits - simply to get a better sense of how well the theory translates into practice.

3) Lemma 1, and perhaps even the Blahut-Arimoto iterations (essentially Lemma 2) would benefit from an illustration.

4) Readers might benefit from an illustration of the main idea early on in the manuscript (this is the main idea that we are going to develop). Perhaps also giving examples of learning targets (and how they could be computed through an optimization) early on would help to set up intuitions.

**Comments:**

 * Does the paper (at some stages) assume that the optimal return is known by construction? Would be good to comment on this (where is this assumed, how is this solved in practice).
 * Section 3.2: How is R_bar computed - how does it differ from V_bar introduced earlier? (don’t they both refer to the expected return?
 * Perhaps it is worth pointing out when connecting learning targets to rate distortion that learning targets can be viewed as lossy compressions of the environment which are useful to train policies that perform well in the environment. Since these (lossy) compressions can potentially be much “simpler” than the full environment, they can be determined more easily (higher sample efficiency when exploring).
 * Line 207: IDS first used, but acronym only spelled out later in the text.
 * A bit nitpicky, but would be nice to discuss (in the conclusions) wow exactly would this help with playing Go on a mobile phone - what is the epistemic uncertainty over the environment? The policy of the opponent? What could a hand-crafted learning target be?

**Time Spent Reviewing:**

3.5

---

### Official Review · Reviewer_6MK2 · 2021-07-16

**Rating:** 7
**Confidence:** 2

**Summary:**

This paper proposes the use of Blahut-Arimoto algorithm to learn an encoder from the source to the target, where the source is the environment and the target the policy to be recovered under a budget of distortion / suboptimality (this budget can be seen as a bounded-rationality). The output of BA algorithm would be a satisficing (good enough) policy, satisfying the budget constraints but also minimizing the mutual information between the policy and the environment. With this paper, the authors propose an improvement over Blahut-Arimoto Thompson Sampling method by using Information Directed Sampling in combination with Blahut-Arimoto algorithm (to compute target action of corresponding source environment). Finally, they provide an additional version based on the variance of the reward as a tractable approximation of the mutual information denominator.

**Limitations And Societal Impact:**

The paper doesn't discuss potential negative societal impact however I think the social impact would be the ones related with the cost of exploration in safe critical scenarios. However, given that this methodology can reason about uncertainty and balance exploration and exploitation accordingly, it can be tuned towards more safe behaviours.

**Main Review:**

A disclaimer here, i'm reading the paper from the perceptive of someone who would be very interested in exploration and value of information but doesn't have much experience in the underlying theory. Having said that, I find the paper very clearly written and have learned a lot while reviewing it - i appreciate that. The suggestions I make will be more on the side of helping readers like me to make it even more accessible.

I understand the idea behind the target (a random variable that partially identifies the environment) but i'd like to see an example of what a target can be.

I find this work related to active learning however this is not discussed in the paper. Specifically, those two papers seem most relevant :
[https://arxiv.org/abs/1112.5745](https://arxiv.org/abs/1112.5745)
[https://arxiv.org/abs/1904.05268](https://arxiv.org/abs/1904.05268) (although they assume observational data i think it can get extended to the interventional case as well)

257: Is \bar{R} the reward of the optimal action? What's the bar semantics? I understand that it's a random variable which depends on the environment but i'd like to see that part of the text (it helped reading this [https://arxiv.org/pdf/2101.06197.pdf](https://arxiv.org/pdf/2101.06197.pdf) but as i mentioned, my suggestion is to make the paper more self contained).

The algorithm, if I understand correctly, relies on the expected regret. However, the optimal reward seems like privileged information, unless the designer makes an assumption about access to this information. Or do you assume to compute through sampling from your belief over the environments which sound expensive but more realistic assumption? I'd like to see this point being clarified.

**Time Spent Reviewing:**

8

---

### Official Review · Reviewer_Mktz · 2021-07-16

**Rating:** 6
**Confidence:** 2

**Summary:**

The paper tackles the problem of learning deliberately suboptimal actions in a bandit setting, with suboptimality being traded off with the amount of information needed from the environment to learn such actions, as formulated in [1]. The main contribution of the work is the combination of the learning target formulation with a learning algorithm based on information directed sampling (BLAIDS), instead of the Thompson sampling algorithm BLAST proposed in [1].

**Limitations And Societal Impact:**

Yes

**Main Review:**

Originality: The paper proposes a novel combination of two previous ideas: the computation of the optimal learning targets trading off optimality and information usage using Blahut-Arimoto as recently introduced in [1], and information directed sampling approaches towards learning the target actions. The proposed approach is a natural extension of [1], replacing the Thompson sampling procedure with an IDS procedure instead. Related work is adequately addressed. While not necessarily a surprising result, I believe the proposed algorithm is a valuable contribution in showing that these two ideas can be combined to provide improvements.

Quality and Significance: The submission appears to be technically sound, with experimental evidence that the proposed BLAIDS algorithm outperforming the earlier Thompson sampling based algorithm BLASTS in synthetic bandit settings while also allowing for tradeoffs of suboptimality vs learning speed (as measured by cumulative regret in earlier time periods). While the experiments do validate BLAIDS as a clear improvement over the earlier BLASTS algorithm, the paper would benefit from more thorough experimentation, for example examining how different configurations of the bandit problems influence the performance and tradeoffs of BLAIDS compared to regular IDS, as well as evaluating how general the improvements provided by using an adaptive $\beta$ are.

Clarity: The paper is well written, providing a clear review of how rate-distortion theory can be related relates to trading off information usage and policy optimality as well as a clear presentation of how to integrate IDS with the optimal learning targets. The experimental results are also clearly presented.

**Time Spent Reviewing:**

4

---

### Official Review · Reviewer_osWU · 2021-08-02

**Rating:** 6
**Confidence:** 3

**Summary:**

In reinforcement learning, the ideal thing to do would be to learn all of the information there is about the environment, and then use that to execute the optimal policy. However, in practice, it is impossible to learn _all_ of the necessary information in very complicated environments. Thus, we instead would like to define a _learning target_ that only incentivizes learning a relevant subset of information that is useful for acquiring reward. The learning target must be chosen such that (a) it does not require too much information to learn, and (b) the optimal policy given only the information in the learning target has low regret (relative to the optimal policy given arbitrary information).

So far learning targets have been coded by hand. Ideally we would instead select an appropriate learning target automatically, that outperforms targets coded by hand. The core insight is that constraint (a) is simply lossy compression, where constraint (b) can be thought of as a constraint on the _distortion_ on regret from throwing away some information in the lossy compression. We can then apply standard approaches from rate distortion theory to choose a target that best fits both constraints.

Once we have selected an appropriate target, we must also choose actions that provide us information about the target. This can be done through _information-directed sampling_, in which we choose actions that minimize the ratio between (expected) regret incurred and information gained about the learning target.

The authors instantiate this framework in the case of multiarmed bandits, where the learning targets are environment-conditional policies $p(a \mid \mathcal{E})$. In the case of zero distortion, the learning target is the optimal environment-conditional policy (which obtains zero regret). As distortion increases, the learning target becomes more and more “smeared” (e.g. in environments where the best and second-best arms are very close in reward, the learning target might put similar probabilities on both arms), making it easier to learn. They show that the resulting algorithm outperforms other alternatives that don’t use automatically generated learning targets and/or information-directed sampling.

**Ethical Concerns:**

No ethical concerns.

**Limitations And Societal Impact:**

As mentioned above, there seems to be a mismatch between the motivation and the actual work done, which I would encourage the authors to fix.

**Main Review:**

The author response made me somewhat more convinced of the benefits of learning targets, and so I have raised my score to a 6.

----

Originality: To my knowledge the idea of automatically constructing learning targets has not previously been combined with the idea of information-directed sampling. However, the method is quite straightforward: it seems to be as simple as taking BLASTS and replacing the Thompson sampling portion with information-directed sampling. In that sense the work seems fairly incremental. (Indeed, many of the points in the paper have substantial overlap with those made in [Arumugam and Van Roy, 2021].)

Quality: To my knowledge the claims made in the paper are correct, though I did not carefully check the math (note that many of the theorems / lemmas come from other papers and so hopefully should be correct.)

Clarity: I found the paper hard to follow due to my lack of background in rate distortion theory. For example, in Section 2.3, the authors simply introduce the rate distortion function, which I initially assumed would be used to quantify distortion in some way, but which actually turns out to be the statement of the constrained optimization problem. (The authors could have mentioned that this was the purpose of the rate distortion function.) More generally, after spending a lot of time understanding the background, the paper made sense to me, but initially I kept getting lost due to small misunderstandings like the one above.

(I would normally not make a big deal out of this, as some amount of background is expected in reading a research paper, but the authors explicitly claim “we do take our contribution to be distilling such key results from the information-theory literature for broader consumption by the sequential decision-making community and, to that end, offer a full presentation for the benefit of readers.”)

Significance: My main critique of this paper is that it doesn’t seem to deliver on its motivation.

From Sections 1-3, my understanding is that the motivation is to produce learning targets that contain less information, so that it is easier for an agent to pick up on the relevant information in complex environments where they cannot possibly learn _all_ of the information. However, as the authors note in lines 200-202, “Moreover, computation of the distortion associated with each possible realization of the target policy would require an independent policy-evaluation step to compute the value function”, which seems like it is incompatible with complex environments (though not necessarily impossible). This is a fundamental issue with the approach (given the motivation) and can’t simply be deferred to future work.

In terms of experimental results, given the motivation, I would like to see an algorithm presented that runs in a complex environment, where the use of an automatically constructed learning target leads to better performance than the default target of the optimal policy, because when using the default target the agent fails to learn the relevant information.

The experiments do show that the automatically constructed learning target leads to better performance, but as far as I can tell, this is not because the agent failed to learn the relevant information when given the default learning target, but because the agent learned _too much_ information when given the default learning target. With both the default and automatically constructed learning target, the agent eventually learns the optimal policy: it just takes longer with the default target, presumably because we select actions that continue gathering information that isn’t relevant, which the automatically constructed target gets rid of (since its entire goal is to get rid of information that doesn’t affect the expected regret much).

So it seems like the actual story in the experiments is “we select actions that prioritize information-gathering too highly; to remedy this we can automatically construct a learning target that removes the irrelevant information, thus preventing our agent from trying to learn it leading to faster convergence”. This is pretty different from the motivation given in the paper (in particular it has nothing to do with how “complex” the environment is).

Of course, I could imagine that the paper had been written with exactly this motivation in mind (the agent tends to acquire too much information, let’s fix that). And indeed it seems like information-directed sampling will sometimes acquire too much information: for example, information-directed sampling will “pay” the same amount of expected regret to distinguish between the best and second-best action, regardless of whether they differ in reward by 0.00001 or by 1, even though in the former case the information hardly matters.

But then I think there would be other approaches which I would like to see comparisons to. Most obviously, can we try to directly fix the problem with information-directed sampling? Information-directed sampling quantifies the _cost_ of acquiring information, but it doesn’t quantify the _benefit_ of that information. Can we instead have valuable-information directed sampling, which chooses actions that minimize the ratio of expected next-step regret to expected decrease in regret in future timesteps? For example, let $\pi^*_{H_t}$ be the policy that makes a best guess about $A^*$ given the history $H_t$, and then always takes that action forever, and let $\overline{V}^{\pi^*_{H_t}}$ be its value. Then we could quantify:

$Cost = \mathbb{E}_{\pi}[\overline{R}(A^*) - \overline{R}(A_t) \mid H_t]$

$Benefit = \mathbb{E}_{A_t \sim \pi, O_t} [ \overline{V}^{\pi^*_H} ] - \overline{V}^{\pi^*_H}$

(For some reason I can’t seem to subscript the $H$ in the above equation. The first one is supposed to be $\overline{V}^{\pi^*_{H_{t+1}}}$ and the second is supposed to be $\overline{V}^{\pi^*_{H_t}}$.)

We could then minimize Cost - Benefit. It seems likely that this will fix the issue where IDS acquires too much information, without requiring the complex machinery from rate distortion theory used in this paper. I would be interested in seeing experimental results from such an approach (using the default learning target of the optimal policy).

**Time Spent Reviewing:**

4

---

### Decision · Program_Chairs · 2021-09-27

**Decision:**

Accept (Poster)

**Comment:**

After discussions between the authors and reviewers, it appears as though the reviewers have all reached a consensus that this paper is worthy of acceptance, so I am happy to support that consensus.

I want to thank the authors for their detailed responses and thank the reviewers for engaging in discussion with the authors.